# Lighting a Path for Autonomous Vehicle Communication: The Effect of Light Projection on the Detection of Reversing Vehicles by Older Adult Pedestrians

**DOI:** 10.3390/ijerph192214700

**Published:** 2022-11-09

**Authors:** Brian Mason, Sridhar Lakshmanan, Pam McAuslan, Marie Waung, Bochen Jia

**Affiliations:** 1College of Engineering and Computer Science, University of Michigan-Dearborn, Dearborn, MI 48128, USA; 2College of Arts, Sciences, and Letters, University of Michigan-Dearborn, Dearborn, MI 48128, USA

**Keywords:** elderly pedestrian, gerontology, parking lot, autonomous vehicle, V2X, automotive lighting, human-centered design, downward gaze, vehicle intent, pedestrian instruction

## Abstract

Pedestrian understanding of driver intent is key to pedestrian safety on the road and in parking lots. With the development of autonomous vehicles (AVs), the human driver will be removed, and with it, the exchange that occurs between drivers and pedestrians (e.g., head nods, hand gestures). One possible solution for augmenting that communication is an array of high-intensity light-emitting diodes (LEDs) to project vehicle-to-pedestrian (V2P) messages on the ground plane behind a reversing vehicle. This would be particularly beneficial to elderly pedestrians, who are at particular risk of being struck by reversing cars in parking lots. Their downward gaze and slower reaction time make them particularly vulnerable. A survey was conducted to generate designs, and a simulator experiment was conducted to measure detection and reaction times. The study found that elderly pedestrians are significantly more likely to detect an additional projected message on the ground than detect the existing brake light alone when walking in a parking lot.

## 1. Introduction

In “Turn Signals are the Facial Expressions of Automobiles”, Don Norman wrote: “social interaction is enhanced when the participants know not only what is happening at the moment, but what will happen. Of all the signals of the automobile, only the turn signals announce intentions”. [1]. The ability of a vehicle to convey intent is vital to the safety of those outside of the vehicle. With the development of AVs, the human driver will be gradually removed, and with it, the exchange that often occurs between vehicle operators and pedestrians (e.g., head nods, eye contact, hand gestures). The lack of vehicle-to-pedestrian (V2P) interaction in the autonomous future may prove to be perilous. AV researchers, engineers, designers and manufacturers are all working to replace these human communications with V2P messages emitted from the vehicle.

This future state of affairs, where pedestrians and drivers cannot communicate, exists in today’s parking lots. Even attentive pedestrians cannot see the driver of a reversing car but instead must infer from the vehicle signals (reverse lights, brake lights) and context (engine noise, doors slamming, car movement, and other pedestrians) to whether or not a car is exiting its parking space. However, pedestrians do not always pay attention [2] or may be otherwise occupied [3]. Additionally, for drivers—even with the benefit of backup cameras and other technology—unsafe reversing and obstructed views increase the risk of a fatal pedestrian crash [4].

Elderly pedestrians are particularly vulnerable [4], as they have longer detection and reaction times [5]. They are more likely than pedestrians of other age groups to be struck by a car in a parking lot, and when struck, they are significantly more likely to sustain a debilitating or catastrophic injury than pedestrians in other age groups [6].

Although a reversing vehicle is much less dangerous than a vehicle traveling forward [4], back-over catastrophes remain a problem for elderly adults [7], who comprise a disproportionately large percentage of pedestrians involved in catastrophic incidents with vehicles [8], e.g., pedestrians aged 75+ years are roughly twice as likely to be hit by a vehicle traveling in reverse than forward [6]. The fact that older pedestrians are particularly vulnerable to being backed over by cars is key to our proposal to locate a high-intensity light projection system in the rear of the vehicle.

Elderly pedestrians may be less likely to see a reversing vehicle for reasons associated with aging: stooped posture, irregular gait, and downward gaze. One study [9] showed that upward gaze impairment becomes apparent among adults in their 70s, and the mean angle of upward gaze was significantly smaller than that of downward gaze. Decreased bone density, hardening of intervertebral discs, and decreased muscle mass can all lead to hyperkyphosis, which occurs when the angle of posterior spinal curvature is beyond healthy ranges [10]. One adverse health effect of the cervical spine issues brought about by aging is an increased chin-brow to vertical angle [11], which can force the gaze downward. Another adverse health effect of cervical hyperkyphosis is a forward shift in the center of gravity [12], which can cause or exacerbate balance issues and can lead to falling and a fear thereof. Falls are a known and prioritized risk in the elderly, particularly as they relate to hip fractures, which are difficult to recover from and may decrease subsequent quality of life [13]. In addition, the focus required to make micro-corrections to shifting balance caused by an unusual gait can distract from the attention required to avoid reversing vehicles in parking lots.

The fact that the effects of aging push the gaze of elderly pedestrians downward, combined with the fact that they are particularly at risk for being struck by reversing vehicles, made us choose this demographic to study the efficacy of projected light messages on the ground behind reversing cars. Specifically, our study quantitatively measures the effectiveness of a proposed vehicle safety feature that projects warnings for pedestrians on the ground behind a reversing vehicle (as shown in Figure 1). Even a small improvement in reaction time could have sizable safety benefits. This experiment examines if the addition of a rear projection to the existing vehicle reverse signaling will increase detection and lower reaction times.

There are existing applications (shown in Figure 2) for on-ground messaging geared toward pedestrians. Those involving projected light are currently relegated to branding or entertainment purposes, as existing technologies cannot yet be relied on for safety-critical applications. As lighting technology rapidly improves, using an exterior lighting apparatus to project messages becomes increasingly viable [14,15,16].

However, parking lots remain the most viable use case for this type of messaging. For the projected image to be useful to pedestrians, the AV would need to be traveling at a speed low enough for the pedestrian to have sufficient time to detect the projected message, determine what it means, and react to it. A projector array can only throw an image so far before becoming unreadable, and thus would give an inattentive pedestrian an extremely short, if not impossible, time to react to the oncoming car.

Although this experiment assumes a projection system so strong as to be seen during the day, any sort of projection system would be more effective at night, where the contrast is higher. Scenarios involving cars striking pedestrians are extremely sensitive to the level of ambient lighting, as catastrophic pedestrian crashes (outside of parking lots) are three to four times more likely in the dark than in the daytime [20]. However, for the parking lot pedestrian, crashes were most likely between 12:00 P.M. and 6:00 P.M., when most commercial businesses are open [6]. Taking into account the time of day, redundant safety systems, and future advances in lighting technology, for purposes of this study, we assume that the projector light will be strong enough to be a visible messaging solution from day to night.

Research has shown that pedestrians would trust AVs more if the vehicles communicated their intent [21] and that a projected illuminated crosswalk is a desirable way to achieve this [22]. However, pedestrian messages take many forms in the world. Some messages originating from vehicles (turn signals, reversing indicators) describe the vehicle’s intent, which the pedestrian must then interpret. Other messages, such as a flashing high beam or a honked horn, do not speak to the intent of the vehicle but rather serve as a missive to the endangered pedestrian. Other messages originating from infrastructure can communicate instructions to vehicles (traffic lights), instructions to pedestrians (cross/do not cross), or both (stop sign). An effective pedestrian message requires clarity of intent and result. When choosing what messages to project behind the car, one can refer to existing research on automotive signage, detection, and familiarity.

Research on variations in signage (larger size, higher contrast or reflectivity) and its impact on driver hazard detection [23] might generalize to pedestrian situations. Some research on road signage has examined how placement, reflectivity, and size of a road sign might attract attention [24,25,26]. In studies where participants were asked to verbally report signage as attracting their attention, detection rates were relatively low (e.g., as few as 10% of the traffic signs present were reported) [25,26]. Furthermore, research on symbols indicates that, in many instances, these are difficult to recognize or are poorly understood, particularly by older drivers [27]. Pedestrian safety might be improved by warning of oncoming vehicles. One study focused on developing visual warning systems for pedestrians through wearable devices [28].

Methods commonly used to access sign recall or recognition include tracking eye movement, eliciting verbal reports from participants, recording participant behavior, and participant recall or recognition of signs at some point later in time [29]. One particular study [30] examined different signs when provided separately, simultaneously, and successively. Researchers found that successive presentation of the same sign in a short spatial interval increased its effectiveness (i.e., a driver registering the presence of the sign). More specifically, younger drivers, professional drivers and drivers who drive more often were better in recalling the signs. In the present study, a group of older adults was asked to indicate their preference for a variety of graphics used to indicate vehicle intent (e.g., graphics that spelled the word “reverse” or the word “back”). They were also asked to determine which graphic (one or two arrows pointing backward, or lines reflecting the path of the tires of a reversing car) best reflected the intended path of the car. This is consistent with assumptions about older people’s desire for familiarity and could have a positive effect on trust-building while AVs are incorporated into the existing motoring ecosystem.

John Tsotsos and his research team at York University (Canada) have made significant contributions to the study of AV–pedestrian interactions. We cite two representative publications that relate to our work the most. In [31], they studied the interaction between AVs and pedestrians in urban environments, specifically crosswalks, and concluded that head orientation is very important in vehicle–pedestrian interactions. In [32], they provided a very comprehensive survey of existing literature on the topic—with a focus on how demographics of the pedestrians, traffic dynamics, and environmental conditions affect this interaction. A review of various design approaches for AVs that communicate with pedestrians, as well as visual perception and reasoning algorithms that are tailored to understand pedestrian intention, is also included in [32].

Elsewhere at the University of Michigan, researchers are also very active in studying AV–pedestrian interactions. Michael Flannagan and his research team’s work are relevant to our study: Ref. [33] is in one of their earliest studies we are aware of relating to the role of vehicle adaptive lighting; the authors of [34,35,36] studied the precise role of light location, intensity and beam spread. Dawn Tilbury and her team have also made significant contributions to this topic: in [37], they measured the behavior of pedestrians at crosswalks using an immersive virtual environment and related it to the real world.

Industry is active in this space as well—using naturalistic driving data, the authors of [38] reported models of driver and pedestrian yielding behavior at crossings; Ref. [39] compared two different crosswalk designs from a pedestrian–vehicle interaction point of view; and [40] provided a survey of published literature on the prediction of pedestrian motion.

Taking these factors into account, this research posits the following: (1) the addition of a rear projection to the existing vehicle reverse signaling will increase both the rate and speed of detection of a vehicle planning to reverse; and (2) participants will prefer existing signage conventions and wording compared to novel signage.

## 2. Survey Materials and Methods

To move toward an ideal projected design (one that is more detectable and leads to a quicker response time), a survey was first conducted to better understand elderly pedestrians’ behaviors, attitudes and V2P messaging preferences. The results were then used to design two projections to be tested against each other. One design would be centered around the idea of “vehicle intent” (e.g., this vehicle is in park, this vehicle is reversing,). The other would be centered around “pedestrian instruction” (e.g., stop walking, get out of the way, be cautious).

### 2.1. Participants

Participants were recruited through Amazon mTurk (a crowdsourcing platform where workers can complete tasks such as answering surveys) to complete a survey that consisted of 70 questions. They were 470 residents of the United States who were over 55 years of age. The majority of respondents were female (62.53%). Respondents indicated their age using these categories: 55–64 (54.68%); 65–74 (40.43%); and 75–84 (4.42%) with one respondent within the 85+ category. A sizable number (41.06%) of respondents reported having difficulty seeing without glasses. Participants also reported hearing deficits (5.27%) and mobility concerns (18.09%, e.g., difficulties walking to and from their cars), with 11.7% of respondents using a walker. Participants had one hour to respond to 70 questions. Data were collected over two weeks.

### 2.2. Measures

The survey items were divided into four parts: *Demographics*, *Pedestrian Behaviors*, *Message Interpretation*, and *Design Preference*. Respondents also had the opportunity to elaborate on their responses. Demographics and Pedestrian Behavior items were meant to better understand the mindset of the American elderly pedestrian, whereas the Message Interpretation and Design Preference items were included to elicit design opinions regarding preferred graphics, wording, and metaphors to develop the two designs that were used in the experimental portion of the study. The two designs were compared in a VR simulator to determine their impact on both overall detection as well as the reaction time to a reversing vehicle.

In addition to standard *Demographics* items, items pertaining to visual and auditory acuity and to physical agility were included. The *Pedestrian Behavior* items focused on specific behaviors that participants engaged in as pedestrians (e.g., waiting for the walk sign when deciding to cross the intersection; waiting for oncoming cars to come to a complete stop at a stop sign before crossing the street).

The *Message Interpretation* items presented participants with pictures of various projected messages and asked them to respond with what they believed was being communicated. Six survey items pertained to two signs (Stop and Yield) in three orientations (as in Figure 3) and two points of view (passing pedestrian and passing driver). Multiple choice options included:(a)I (the pedestrian or driver) am supposed to follow the sign’s instructions;(b)Other drivers are supposed to follow the sign’s instructions;(c)The vehicle is currently following the sign’s instructions;(d)Both vehicles and pedestrians should follow the sign’s instructions.

Initial *Visual Design Preference* items were a series of ten A vs. B questions about what types of visual cue (text, icon, color, etc.) best communicate an idea about pedestrians in a parking lot (I should stop walking, this car is backing up, etc.) as in Figure 4. The projection designs were generated based on best road signage practices [27,41,42].

The final task of the *Visual Design Preference* items was to narrow down the proposed designs for both categories. This was achieved in a bracket of sorts, with participants selecting preferred designs within divisions (signage, symbols, wording) and then selecting their favorite of the division winners, giving us both the preferred design and division (mode) of the message. To determine respondents’ preferred graphic to represent *Pedestrian Instruction*, they were asked to choose which projected message they preferred if an autonomous vehicle was “leaving a parking space” and was going to “guide them to safety”. The questions were worded in that way so as not to describe the vehicle’s intent (wanted to back up), but rather the state (leaving the parking space) of the vehicle and what it wanted to communicate to the pedestrian. To determine participant’s preferred graphic to represent *Vehicle Intent*, they were asked to choose which graphic they preferred if an autonomous vehicle “intends to pull out of its parking space” and wanted to communicate that. The question required less delicate wording than the pedestrian instruction, as the vehicle’s intent to reverse was no secret. The “divisions” for Vehicle Intent were verbiage and vectors.

## 3. Survey Results

### 3.1. Message Interpretation

For the interpretation questions, in all six cases, regardless of orientation or signage, most participants believed the projected message to mean “both pedestrians and cars around the blue car” should follow the projected signage. When the participant was asked to imagine themselves as the pedestrian, they were more likely to believe the message was for pedestrians, and when asked to imagine themselves as a driver, they were more likely to believe that the message was for drivers. Signs with text oriented parallel to the rear bumper were thought to be describing the state of the car behind it.

### 3.2. Visual Design Preference

Participants showed a preference toward existing signage conventions and wording. Attempts to being clever, such as using “back” to suggest both vehicle and pedestrian should be moving back, were soundly rejected. There was a preference that graphics run perpendicular to the rear bumper of the car, as the messages were more clearly understood to be for pedestrians/motorists and not describing the state of the vehicle.

Participants preferred stop signs, the color red, and combinations of symbols and words. Also of interest is the preference for the yellow yield sign over the red yield sign, as the United States moved from yellow signs to red signs in 1971 [43]. The more elderly participant base may account for this response.

The final projection designs (shown in Figure 5 and Figure 6) were each a combination of the two highest vote getters from their respective survey sections (*Pedestrian Instruction* and *Vehicle Intent*). Results from the Visual Preference survey were taken into consideration when determining the best way to combine the designs. The *Pedestrian Instruction* projected message, called “Stop Signs”, combined the winners of the two highest vote getters from the survey into a single concept that incorporated both instructions (double stop), signage (stop signs with text), and the dashed lines (the area to be avoided) that survey participants preferred. The *Vehicle Intent* projected message, called “Reversing Arrow”, combined both vectors (the large arrow suggesting the path of the car) and written instruction (reversing). These would be the designs tested against each other in the VR simulator.

## 4. Simulation Materials and Methods

The goal of the VR Simulation experiment was to test elderly pedestrians’ reaction times to different reverse indications. We aimed to discover if elderly pedestrians are more likely to detect, and quicker to react to, a projected message on the ground than the existing brake light assembly. Reaction times for the existing brake light assembly and the respective projected messages were assessed.

### 4.1. Participants

Simulation participants were 32 seniors (62.5% female), with 27 from a west Michigan Senior Living Community and 5 from an east Michigan Senior Center. Recruitment was restricted to ambulatory participants 65 and older that had walked, with or without the help of a mobility device, in a parking lot within the last month. Almost half were between the ages of 85 and 94 (n = 15; 46.9%), with the remaining participants aged 75 to 84 (n = 9; 28.1%), 65 to 74 (n = 6; 18.8%), with one participant aged 55 to 64 (3.1%) and one over 94 years of age (3.1%). Many participants reported mobility (n = 5; 15.6%), vision (n = 26; 81.3%) or hearing impairments (n = 26; 81.3%). Two cases were removed from the dataset because one participant had vertigo and could not complete the task, and the other was missing substantial data such that a meaningful average reaction time for any of the three conditions could not be calculated.

### 4.2. Materials

The simulator experiments were conducted in a small multi-purpose room at the Senior Center (Figure 7) and in a resident’s room at the Senior Living Community (Figure 8). Participants were instructed to watch the videos being projected onto the wall. A video projector was positioned approximately 143″ from the wall, which led to a projected image 60″ tall with a 122″ diagonal. The bottom of the projection was meant to be as close to the floor as possible to suggest ground level. Participants were seated near a table as close to the projector as the room allowed and were given a keyboard to hold in either their lap or on the desk in front of them. The room was kept as dark as possible, and the test administrator was masked as per COVID safety protocols.

#### Videos

The simulator test used 30 videos that each simulated the gait and gaze of an elderly pedestrian walking through a parking lot. Each thirty-second video would feature the first-person point of view of a pedestrian walking down a row of cars as shown in Figure 9.

At some point in the video, one of the cars would—or would not—shift into reverse and would—or would not—project a graphic behind it. The car in question (the “mover) would either show (a) brake lights only, (b) the Stop Signs projection, or (c) the Reversing Arrow (see Figure 10). In some cases, the car would not shift into reverse at all.

Variables were split up so that projected messages were equally distributed among the videos. Of the 30 videos, one third featured the Stop Signs projection, another third featured the Reversing Arrows projection, and the final third used only the existing brake lights. One video for each direction and parking configuration (six in total) had no cars in reverse. Half of the videos featured the virtual pedestrian walking north, the other half walking south. Half of the videos featured the mover on the far left, the other half featured the mover on the right, and so on. Six of the videos were mirror images of another. The videos had no sound to isolate the visual component.

Designs were digitally added to pedestrian POV videos to appear projected from behind a car. The videos were made using After Effects (2020, Adobe Inc., San Jose, CA, USA, AfterEffects 2020). A .png image containing the image was placed on a ground plane using a 3D tracker camera. The Layer Blending was set to “Color” at 100% Opacity, which “creates a result color with the luminance of the base color and hue and saturation of the blend color. This preserves the gray levels in the image and is useful for…tinting color images”. (2021, Adobe Inc., San Jose, CA, USA, Adobe, 2021) This was chosen to retain the integrity of the image, reflect the texture of the street below, and to give the projected message a realistic relationship with shadow. This approximates a more advanced, but plausibly performing, lighting system than what is currently available.

The parking lot videos were filmed on one-way aisles, as participants would be exposed to double the number of taillights as they would on two-way aisles, and prior studies showed that aisle directions have no effect on crash frequency or severity [6]. When filming, the camera was positioned 60 inches above the ground (approximately the mean standing eye height of Australian men 65 and older) [44], which was used as a proxy for the height of the elderly American pedestrian. Although Americans are generally taller than other populations worldwide [44], there are no nationwide anthropometric data on Americans [45].

### 4.3. Measures

Participants were first asked a series of questions (from the survey) regarding demographic information and pedestrian behaviors. They were then shown a sample video and were pointed out the location of the reversing lamps and explained the concept of the ground plane projection. They were instructed to press pause (spacebar) on a keyboard in their lap when they noticed one of the cars shifted into reverse. Participants reacted to one sample video for practice and were informed that they could quit at any time if they started to feel nauseous.

The 30 videos were shown to the 30 participants. The noted times in the video are the moment the brakes are applied (brake lights on) and the moment the car shifts into reverse (reverse lights on). Times were recorded in hundredths of a second. From this, we can calculate the reaction time to the respective projected message. A “projection only” configuration was never tested, as the ground plane projection would always be used in conjunction with other modalities.

## 5. Simulation Results

Initially, we examined the participant responses for each of the videos to consider when or if the participants indicated that they believed that the car was about to reverse. This resulted in four different types of data points that were not included in the reaction time calculations that follow.

### Main Analysis

Across each of the three conditions, it was common for participants to react to the brake lights (prior to the reverse lights/projection turning on). This tendency did not differ across conditions, *F*(2,20) = 0.21, *p* = 0.81, *η*^2^= 0.02. Similarly, across conditions, there were some videos where participants did not react at all, although this was significantly more likely in the no projection condition, *F*(2,20) = 5.68, *p* < 0.05, *η*^2^ = 0.36. Some participants reacted in anticipation of the car signaling that it was going to begin to reverse, but this did not significantly differ across conditions, *F*(2,20) = 0.33, *p* = 0.73, *η*^2^ = 0.03. Finally, as a result of either experimenter error, participant error, or equipment malfunction, there were some missing data, but this also did not differ across conditions, *F*(2,20) = 0.49, *p* = 0.62, *η*^2^ = 0.05. Table 1. Simulator Study Mean and SD provides the mean and standard deviation related to the number of cases of each type of data that was not included in the reaction time calculations that follow.

At this point, the reaction times for each video within each condition were examined. Some of these were significantly inflated by reaction times that were significantly longer than the majority of times (e.g., responding after the signal had been on for over 20 s). Outliers were identified by examining skewness with all outliers three standard deviations or more from the mean removed for the purpose of analyses. At this point, the average amount of time that it took participants to react following the reverse/projection signal was computed for each of the three conditions. Participants were required to have at least two valid data points per condition to be included in the analysis (n = 18).

A repeated measures ANOVA was conducted to compare the reaction time across the three conditions. There was a significant difference between the conditions, Wilk’s Lambda (2, 16) = 3.95, *p* < 0.05, partial *η*^2^ = 0.33. The reaction time for the reverse signal only condition (*M* = 2.62; *SD* = 0.38) was significantly longer than either the reverse signal + stop projection (*M* = 1.82; *SD* = 0.19) or the reverse signal + arrow projection (*M* = 1.45; *SD* = 0.14) conditions. The difference between the stop and arrow projection conditions was marginally significant (*p* = 0.06).

## 6. Discussion

Although the test results showed that the difference between the two proposed designs was not statistically significant, if given the opportunity, a good message design is likely a combination of a linear directional element indicating the path of the vehicle and instructional road signs or verbiage. According to the survey, participants preferred the stop sign to the yield sign and reported to stop for reversing cars, but this led to questions about the wisdom of a private vehicle commanding a pedestrian to stop. For that reason, something akin to Figure 11 would be recommended as the optimal projected message for a reversing AV. Any design within the parameters of what is currently recognizable road symbology will likely have a positive effect on safety.

Many participants said they looked for motion when scanning a parking lot. It is likely that if the lines moved (perhaps following the intended path of the vehicle), that detection would increase. This should not be confused by a flashing projection, because while younger populations are more likely to notice a flashing light, the same is not true for the elderly [46].

The study found that elderly pedestrians are significantly more likely to detect a projected message on the ground than detect the existing brake light when confronted with a vehicle in reverse while walking in a parking lot. In addition, the participants reacted faster to the projected messages than the brake lights alone. Thus, having projected images in addition to the standard reverse signal increased the frequency of elderly pedestrians detecting reversing vehicles and decreased the time it takes them to react.

### 6.1. Limitations

The largest limitation of this study is the simulator. A static, seated simulator was selected to give elderly participants the highest level of comfort and to minimize chances of vertigo. Future experiments could involve different postures, environments, movements, and designs. There were significant safety concerns about setting up this type of experiment in a real parking lot with real (potentially) reversing vehicles. Accuracy could be increased by evolving the simulator or by creating a prototype. An indoor, controlled prototype could be used to achieve accurate eye tracking and distance data, but the element of surprise and the context of a parking lot would be lost. A more immersive VR-goggle experience could also present high risk to seniors due to VR motion sickness and balance issues.

### 6.2. Future Research

Future work could increase both the accuracy and scope of the experiment. The accuracy of the study could be greatly increased with the addition of eye tracking and a larger, more immersive simulator. In addition, having the elderly pedestrian perform the tasks in the standing position would give an additional layer of realism, but the tasks would have to be shortened as a courtesy to the participants.

Testing the projected messages in other contexts (nighttime, wintertime, covered parking) and in other populations may prove fruitful. While this experiment simulated a daytime walk, running the experiment in darker conditions would most likely improve the performance of the projections due to the increase in contrast.

Testing projected messages on participants between 15 and 19 years of age—for whom the rate of pedestrian crashes in parking lots is highest [6]—could also greatly increase the safety of a vulnerable population. This could also be expanded to younger children, for whom collisions result in the highest rate of catastrophe. The projector would also need to be tested on the front of the car, as other age groups are not as uniquely imperiled by back-over accidents. Another potential advantage of the added projection is to grab the attention of a distracted pedestrian looking at their phone. Pedestrians that are texting on their phone are 3.9 times more likely to exhibit dangerous crossing behavior than those that are not distracted [3].

There are also larger questions about the solution. Would ground plane projection perform better as part of the infrastructure rather than an AV? Would it lessen or exacerbate potential liability issues? Does the level of autonomy, or perceived autonomy, of the vehicle, affect the detection and reaction of the projected messages?

## 7. Conclusions

When walking in a parking lot, elderly pedestrians are significantly more likely to detect a projected message on the ground than detect the existing brake light configuration. By adding a projected message designed with a combination of vehicle intent and pedestrian instruction, we can increase detection of and decrease reaction time to reversing vehicles and make elderly pedestrians safer.

## Figures and Tables

**Figure 1 ijerph-19-14700-f001:**
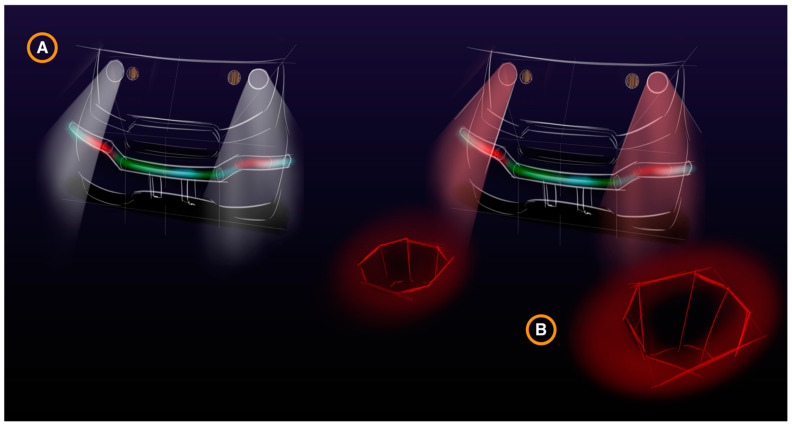
AV Projection Concepts by Brian Mason for MDAS.ai, an autonomous shuttle being developed from the ground-up for use within University of Michigan–Dearborn. (**A**) Shows the standard OEM headlights and lightbar indicator; (**B**) shows the projected warning. In this design, they are meant to simulate a hole in the shape of stop signs.

**Figure 2 ijerph-19-14700-f002:**
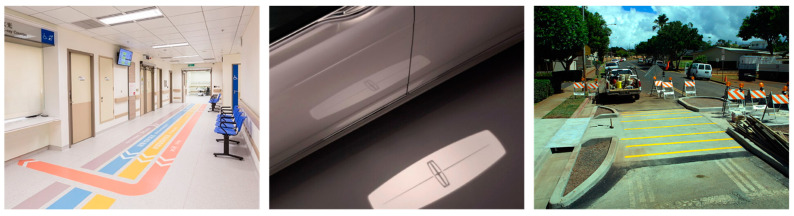
(**Left** to **right**) Tin Shiu Wai Hospital floor wayfinding [17], Lincoln Welcome Lighting [18], and a raised crosswalk [19].

**Figure 3 ijerph-19-14700-f003:**
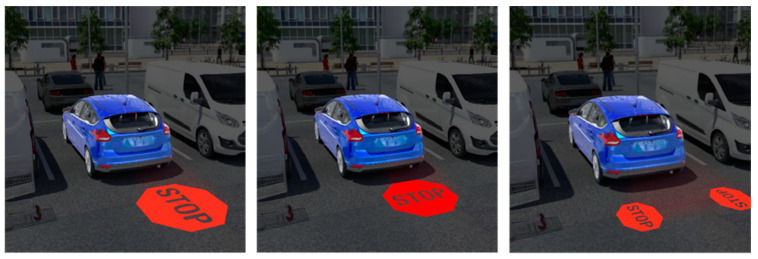
Graphics from three *Message Interpretation* items. Each item’s question read: “You are a pedestrian walking through a parking lot. An autonomous car is projecting this graphic behind it. What does the graphic mean to you?”.

**Figure 4 ijerph-19-14700-f004:**
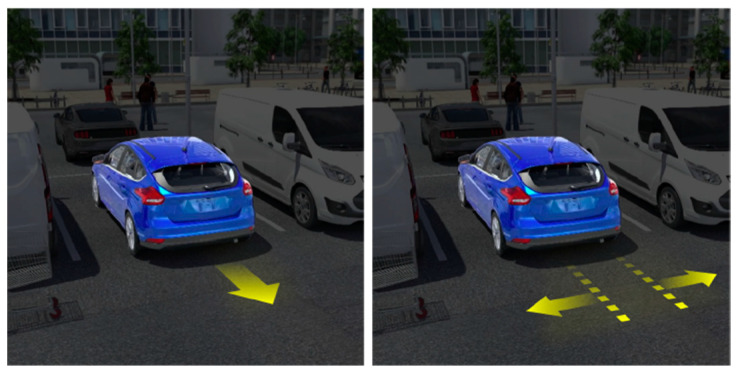
Graphics from a *Visual Design Preference* item. Each item’s question read: “You are a pedestrian walking through a parking lot. A parked autonomous car has shifted into reverse. Which projected graphic do you think communicates this best?”.

**Figure 5 ijerph-19-14700-f005:**
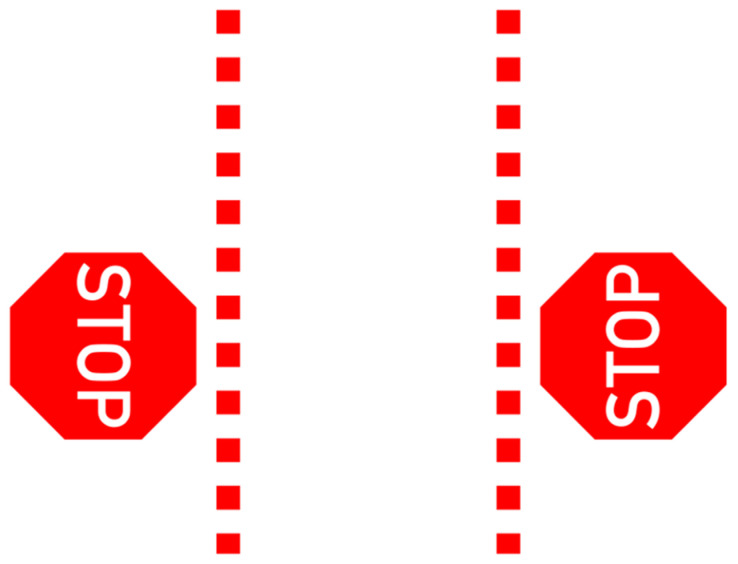
Projected graphic representing “Pedestrian Instruction”, hereafter referred to as “Stop Signs”.

**Figure 6 ijerph-19-14700-f006:**
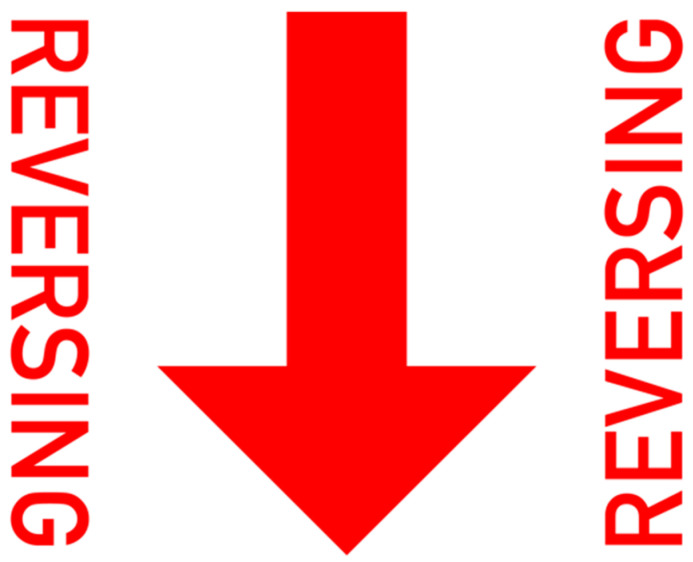
Projected graphic representing “Vehicle Intent”, hereafter referred to as “Reversing Arrow”.

**Figure 7 ijerph-19-14700-f007:**
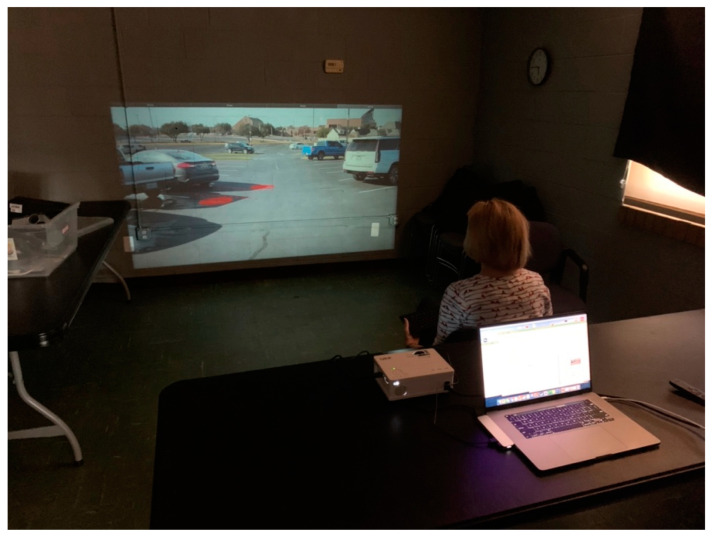
Experiment room in Pittsfield Senior Center.

**Figure 8 ijerph-19-14700-f008:**
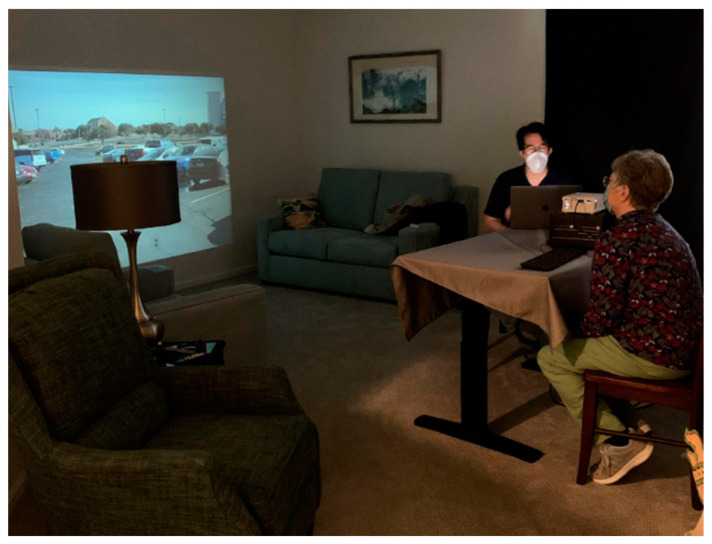
Experiment room in Freedom Village Senior Living Center.

**Figure 9 ijerph-19-14700-f009:**
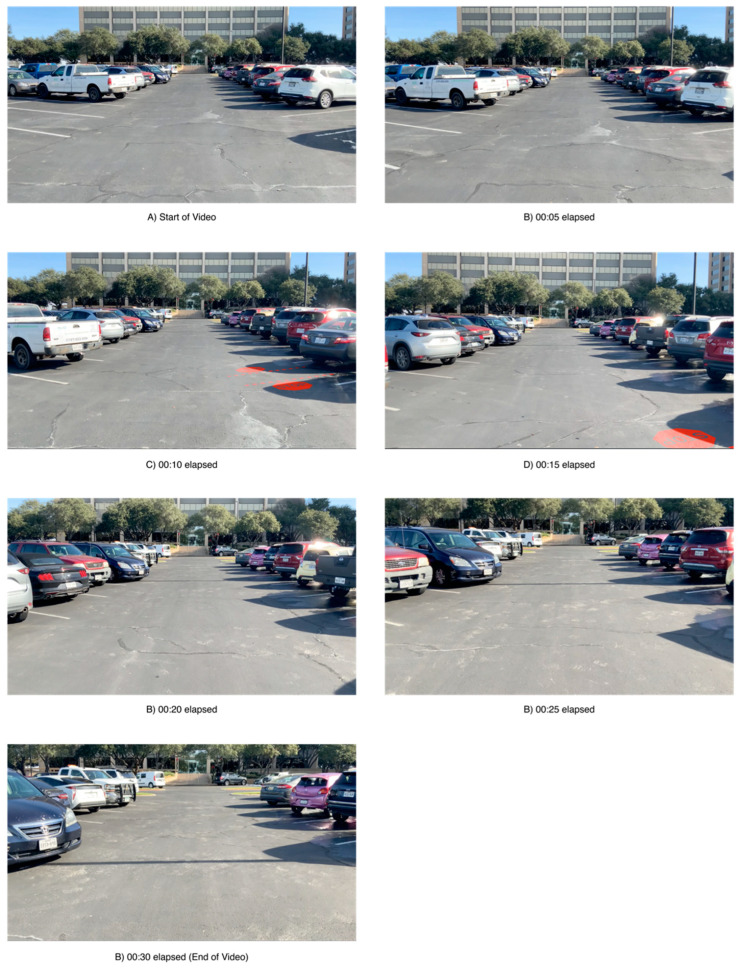
Example time lapse of simulator video with projected message.

**Figure 10 ijerph-19-14700-f010:**
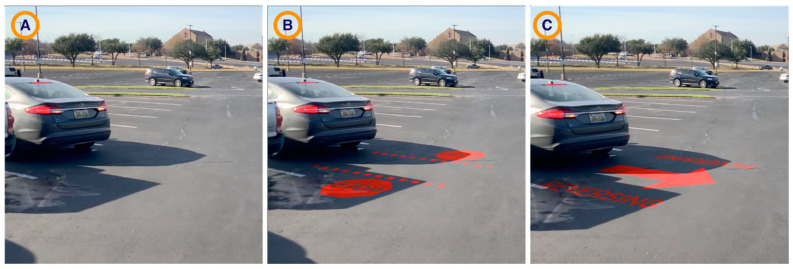
Projected messages in simulator video: (**A**) Brake Light Only, (**B**) Stop Signs (Pedestrian Instruction), (**C**) Reversing Arrows (Vehicle Intent).

**Figure 11 ijerph-19-14700-f011:**
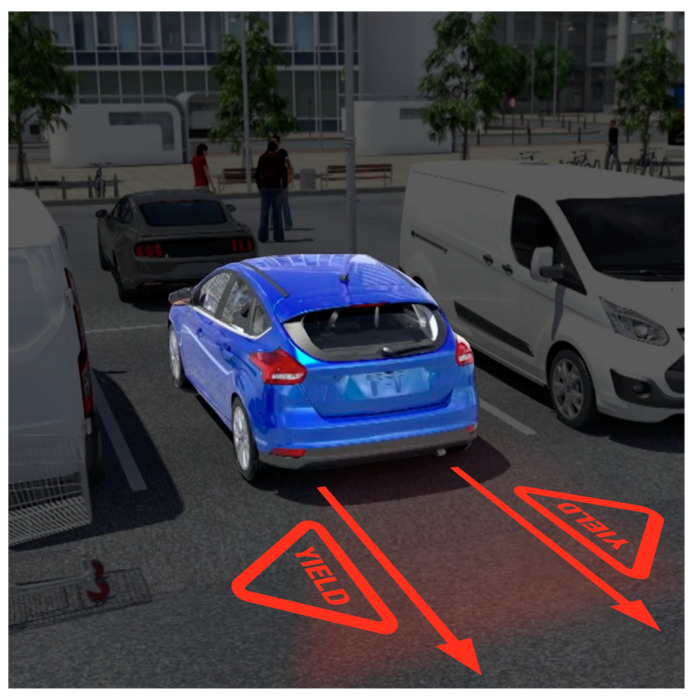
Projected image final design recommendation.

**Table 1 ijerph-19-14700-t001:** Simulator Study Mean, SD, and number of cases by condition.

		Condition	
Title 1	No Projection (n = 8)	Stop Projection (n = 7)	Arrow Projection (n = 8)
Reacted to brake lights	2.13 (2.90)	1.29 (1.70)	1.75 (2.90)
Did not react at all	15.13 (8.82)	6.00 (6.24)	4.05 (4.47)
Anticipatory reaction	2.50 (2.45)	2.71 (2.69)	3.63 (3.54)
Missing data	1.88 (2.64)	1.00 (1.00)	1.63 (0.92)

## Data Availability

Not applicable.

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
