# Peer review of "Lighting a Path for Autonomous Vehicle Communication: The Effect of Light Projection on the Detection of Reversing Vehicles by Older Adult Pedestrians"

_ijerph, 2022, doi:10.3390/ijerph192214700_

Round 1

Reviewer 1 Report

The authors evaluate different lighting message designs for a vehicle backup warning to be displayed on pavement, targeted specifically to older pedestrians. V2P communication is an extremely fertile research topic, especially as AVs currently have limited ability to communicate simple messages to pedestrians. Any attempt to formalize the non-verbal driver-to-pedestrian communication is always appreciated.

The authors lead with a thorough literature review. They make a strong case for targeting older pedestrians. The rationale suffers a bit, though, as the authors acknowledge that current lighting technology cannot project in daylight. This is a significant problem in parking lots where most crashes occur, as the authors correctly note. This is also a problem for older pedestrians, who I assume are generally out during daylight hours and are therefore most at risk during these times. An intervention that can’t help its target demographic is a problem, and the authors need better justification.

A bigger problem is that the paper is framed around using on-pavement lighting for automated vehicles. While AVs need to communicate with pedestrians, their communication is almost entirely about letting pedestrians know when the are safe. They need to let pedestrians know it’s safe to walk, so that pedestrians feel secure and are not unnecessarily delayed. Alerts for when pedestrians are unsafe shouldn’t be necessary at all, as AVs should be programmed so that they are physically unable to hit pedestrians, especially in a low-speed parking lot environment. If a vehicle detects a pedestrian behind it, the AV should simply not move. I cannot envision a scenario where it would have to warn a pedestrian about the AV’s own unsafe behavior. This needs to be addressed. Perhaps the warning is to be used only in situations where the pedestrian is at no risk, and is merely an additional warning to let pedestrians feel more comfortable? But this doesn’t seem to be the authors’ intent, as they speak of actual, not potential, back-over accidents (e.g., lines 441-453).

Is table 1 showing reaction times in seconds? If so, please label accordingly.

The discussion of outlier removal (line 388) is weak. Use a formal outlier removal standard, or provide a visual of all data so that outliers can be seen.

Last sentence of the paper (lines 457-459) seems incomplete.

Reviewer 2 Report

In the simulation part, the following problems need to be further clarified:

1. Is the selected elderly group  as the pedestrian representative(Almost half were between the ages of 85 and 94 , with the remaining participants aged 75 to 84 , 65 to 74 , with one participant aged 55 to 64 and one over 94 years of age) consistent with the elderly group in the actual traffic participants?

2. Section 4.3 is missing.

3. The effectiveness of simulating the test in action through the static test of the testee needs further discussion.

4. The angle of view of the tested sitting posture and standing posture is quite different, and the experiment needs to be further refined.
